# An Evaluation of Two Decades of Aerosol Optical Depth Retrievals from MODIS over Australia

Marie Shaylor [1,2,*] , Helen Brindley [2,3] and Alistair Sellar [4]

1   Science and Solutions for a Changing Planet DTP, Imperial College London, London SW7 2BX, UK
2   Space and Atmospheric Physics Group, The Department of Physics, Imperial College London, London SW7 2BX, UK; h.brindley@imperial.ac.uk
3   National Centre for Earth Observation, London SW7 2BX, UK
4   Met Office, FitzRoy Road, Exeter EX1 3PB, UK; alistair.sellar@metoffice.gov.uk
*   Correspondence: m.shaylor19@imperial.ac.uk

**Abstract:** We present an evaluation of Aerosol Optical Depth (AOD) retrievals from the Moderate Resolution Imaging Spectroradiometer (MODIS) over Australia covering the period 2001–2020. We focus on retrievals from the Deep Blue (DB) and Multi-Angle Implementation of Atmospheric Correction (MAIAC) algorithms, showing how these compare to one another in time and space. We further employ speciated AOD estimates from Copernicus Atmospheric Monitoring Service (CAMS) reanalyses to help diagnose aerosol types and hence sources. Considering Australia as a whole, monthly mean AODs show similar temporal behaviour, with a well-defined seasonal peak in the Austral summer. However, excepting periods of intense biomass burning activity, MAIAC values are systematically higher than their DB counterparts by, on average, 50%. Decomposing into seasonal maps, the patterns of behaviour show distinct differences, with DB showing a larger dynamic range in AOD, with markedly higher AODs ($\Delta$AOD$\sim$0.1) in northern and southeastern regions during Austral winter and summer. This is counter-balanced by typically smaller DB values across the Australian interior. Site level comparisons with all available level 2 AOD data from Australian Aerosol Robotic Network (AERONET) sites operational during the study period show that MAIAC tends to marginally outperform DB in terms of correlation ($R_{MAIAC}$ = 0.71, $R_{DB}$ = 0.65) and root-mean-square error ($RMSE_{MAIAC}$ = 0.065, $RMSE_{DB}$ = 0.072). To probe this behaviour further, we classify the sites according to the predominant surface type within a 25 km radius. This analysis shows that MAIAC's advantage is retained across all surface types for R and all but one for RMSE. For this surface type (Bare, comprising just 1.2% of Australia) the performance of both algorithms is relatively poor, ($R_{MAIAC}$ = 0.403, $R_{DB}$ = 0.332).

**Keywords:** AOD; MODIS; MAIAC; DB; aerosol; Australia; optical depth; CAMS; AERONET

## 1. Introduction

Aerosols are tiny suspended solid or liquid particles in the atmosphere comprised of substances such as smoke, ash, dust, pollution, sulphates and numerous other natural or anthropogenic species. Aerosols affect the climate by altering the Earth's radiation budget through several mechanisms. They absorb and scatter incoming solar and outgoing longwave radiation, alter cloud microphysical properties such as droplet size and concentration by acting as condensation nuclei and even affect the carbon cycle via land and ocean fertilization, through the deposition of nutrients such as iron and phosphorus (for example, in nutrient rich dust) [1–3]. Additionally, particulate matter smaller than 2.5 microns, PM2.5, is known to have negative long term effects on human health and well-being [4]. Decreases in visibility caused by accumulation of aerosols regionally can also negatively impact the local environment [5]. It is clear, therefore, that there is a need to monitor the distribution and evolution of aerosols and to properly assess the accuracy of aerosol observations.

Australia is a growing area of interest in the aerosol remote sensing community. A wide variety of aerosols have been identified over Australia, including dust, smoke, urban pollution, sulphates and sea salt [6]. Australian smoke and dust aerosols are significant contributors to the global budgets of these aerosol types. The region has been estimated to contribute 7% to global fire emissions (1997–2009), making it the fifth largest fire emissions source region globally [7]. Bushfires and wildfires are common along the north and east coasts over grassland and savanna-dominated areas, particularly in the fire season [7]. It is also thought to be the most prominent source of dust in the Southern Hemisphere, making up around 4% of the global dust budget [8]. This is largely due to Australia being host to many ephemeral lakes and other water bodies, some of which are active dust sources year long and which have been estimated to be responsible for around 75% of dust emissions in the country [8]. Nonetheless, Australia remains a relatively understudied region in the literature, and often discrepancies are found when determining spatial/seasonal Aerosol Optical Depth (AOD) trends over the area. Work by Mulcahy et al. [9] shows that the spatial distribution of satellite retrieved AOD over Australia varies significantly depending on which satellite is used for the retrievals. The largest discrepancies are seen in the Austral winter, in which the three satellite AOD retrievals they explore predict levels ranging between 0 and 0.3 over the majority of the land surface. It is clear, therefore, that an accuracy assessment of satellite AOD retrievals from Australia is required.

Arguably, the most popular satellite instrument used for AOD retrieval in the atmospheric community is the Moderate Resolution Imaging Spectroradiometer (MODIS) instrument on board the twin Terra and Aqua satellites. These satellites have a near polar orbiting scan track and a global data record spanning more than 20 years (December 1999–present), achieving global coverage every 1–2 days. MODIS measures spectral radiance of the Earth in 36 bands across the visible, UV and IR parts of the spectrum (0.4–14.4 μm) and uses some of these to perform AOD retrievals. At the time of writing, there are three operational AOD retrieval algorithms applied to MODIS radiances, and these are: Multi-Angle Implementation of Atmospheric Corrections (MAIAC), Deep Blue (DB) and Dark Target (DT). As the name suggests, the DT algorithm was designed to retrieve aerosol over surfaces that appear dark at solar wavelengths, such as the ocean (under non-glint conditions) and dense vegetation. DB and MAIAC were specifically developed to counteract some of the issues encountered by the DT approach over brighter surfaces. Because the vast majority of the Australian land surface is highly reflective, this work focuses on AOD retrievals from only the DB and MAIAC algorithms.

The DB algorithm is one of the most extensively validated AOD retrieval algorithms, and with a wealth of both global and regional validations already performed, its limitations are well characterised. Recent work by Yang et al. [6] and Che et al. [10] explores the performance of the DB algorithm over Australia using Aerosol Robotic Network (AERONET) ground based sun photometers for ground truthing. Yang et al. [6] use DB retrievals to analyse the spatio-temporal distributions of AOD over Australia from 2002 to 2020, overall and by region, with data from the Aqua satellite. They find aerosol levels that peak in the Austral spring and summer and an overall decreasing trend in yearly averaged AOD from the DB Aqua data ($-0.0003$ yr$^{-1}$). They also analyse AERONET records through a yearly time series analysis, noting both increasing and decreasing AOD trends in the long term over specific sites. With an averaged monthly analysis of AERONET records, they similarly find spring and summer peaks at all sites. They go on to further discriminate aerosol properties over Australia, specifying the AOD types present. Che et al. [10] also use AERONET to perform an overall evaluation of DB AOD retrievals over Australia as part of their research, before going on to compare the DB retrievals to an AOD reanalysis product. They find good agreement between DB Terra and Aqua AODs with the AERONET data, with 66% and 67% of points falling within the Expected Error (EE) envelope, respectively, and similar Root Mean Square Error (RMSE) values (0.07 and 0.08). However, coefficients of 0.56 for Terra and 0.68 for Aqua indicate a marginal improvement in agreement for Aqua.

The work presented here builds on these foundations by comparing the DB algorithm with the newer MAIAC algorithm. The MAIAC algorithm has improved spatial resolution, and regional studies have found it to have an enhanced ability over DB to retrieve fine scale AOD features, such as smoke plumes, as well as performing well over complex geographical landscapes [11]. Since the MAIAC algorithm became operational in 2018, there has been one global [12] and numerous regional evaluations of its performance and limitations, including North America [13], South America [14], Asia [11,15–19] and Moscow City [20]. Falah et al. [21] also use AERONET to assess the validity of MAIAC retrievals over North Africa (Algeria, Morocco and Tunisia), California and Germany, to elucidate the effect of different environments (aerosol types, surface properties) on the performance of the MAIAC algorithm. These studies have generally found MAIAC to have good agreement to ground truth sites, often exceeding the performance of the DB algorithm. Whilst the global performance analysis of MAIAC by Qin et al. [12] includes a brief assessment of Oceania aerosol, providing Correlation Coefficient (R), RMSE, mean absolute difference and mean absolute error for four AERONET sites, most of the analysis focuses on global patterns as a whole. They find good performance of MAIAC AOD over Oceania from the AERONET sites used. They go on to evaluate global performance as a function of surface type, finding that MAIAC performs best over highly vegetated surfaces and worst on croplands and barren lands. In this work, we expand on this by taking a detailed look into the performance of MAIAC over Australia, including an assessment of the seasonality of AOD and the effect of aerosol loading and underlying surface type on retrieval performance in the region. To our knowledge, this is the first detailed regional evaluation of the MAIAC AOD retrieval algorithm over Australia.

We compare AOD retrievals over Australia from MAIAC and DB against one another and against AERONET for the period 2001–2020. This article is set out as follows: Section 2 outlines the materials and methods used, including a description of the study area and the MODIS and AERONET data. It also describes both the surface type classification system and methods used to co-locate MODIS and AERONET AOD retrievals. Section 3 presents the results of spatio-temporal trends and seasonal distributions of AOD over Australia, subsequently using the CAMS reanalysis dataset to understand aerosol species' distributions. This is followed by the results of the co-location to AERONET, first for the whole region, then as a function of aerosol loading and finally as a function of underlying surface type. Conclusions from this work are presented in Section 5.

## 2. Materials and Methods

### 2.1. Study Area

Australia is the most prominent source of dust in the Southern Hemisphere, being home to a number of evaporated or ephemeral water bodies which can continuously produce dust storms that can travel far across the country [8]. Prone also to frequent and large bush fires, particularly in the summer months, Australia is additionally a significant contributor to global smoke aerosol [7]. Fires are most prevalent in areas where fuels are more widely available, such as in the forest and savanna covered lands near the north and east coasts. Urban aerosols may be prevalent around urban centres and sea salt aerosols may be prevalent along the coastlines [6].

Australia has several distinct climatological zones (Figure 1). Almost all of the interior is covered by arid deserts, which are largely made up of dune fields [22]. Tropical and subtropical temperate climates dominate to the north and east of the country, experiencing characteristic heavy rainfalls during the monsoon season (November–April) and humid heat in the dry seasons (May–October) [23]. Coastal south and east areas have a warm and cool temperate climate, similar to European climates with 4 distinct seasons. This is also where the majority of the population lives. Arid desert and steppe climates cover the largest fraction of land and are the areas where it is expected that dust activity will be the highest [22]. The analysis performed in this work focuses on all Australian land mass area (including Australian islands such as Tasmania), as shown in Figure 1.

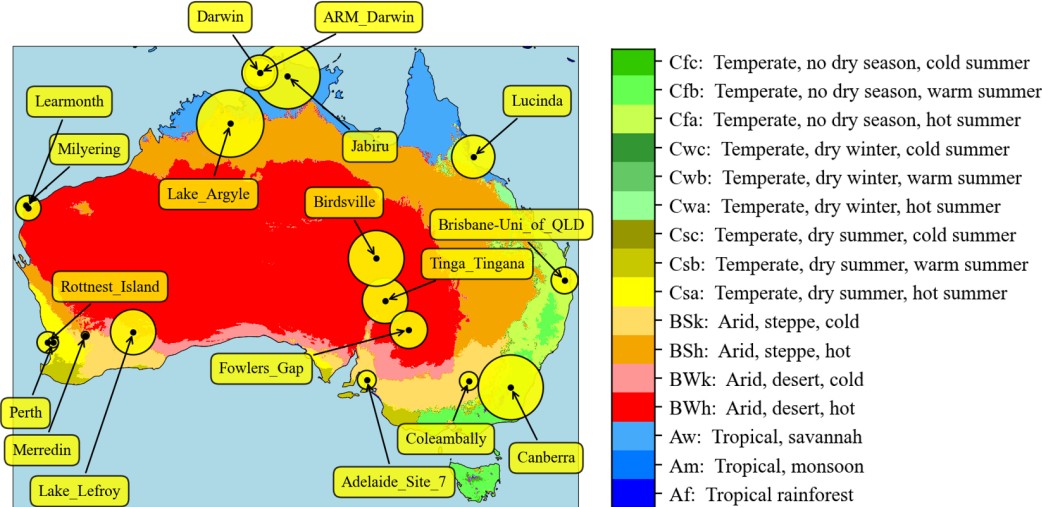

**Figure 1.** Köppen Climate Zones in Australia generated using Copernicus Atmosphere Monitoring Service information (2022) [24]. Overlain are the locations of the Aerosol Robotic Network (AERONET) sites used in this study (see Section 2.2.3). The size of the circle markers are proportional to the length of the data record available. The colour scheme used is taken from Beck et al. [24].

### 2.2. Data Sources

#### 2.2.1. MODIS: MAIAC AOD (MCD19A2)

The MAIAC algorithm [25] is applied to the MODIS instruments on board the Terra and Aqua satellites. The MAIAC algorithm derives AOD at 550 nm from raw MODIS radiance observations by using a dynamical times series approach and image-based processing, which helps to separate out the Earth's surface and atmospheric contributions to the total observed reflectance. In principle, this method allows MAIAC to retrieve AOD effectively over both bright and dark surfaces, for the entire data record (2000–present) [25].

MAIAC data are presented at 1km resolution on a sinusoidal grid and are split into equally spaced 1200 km by 1200 km tiles. Australia falls within the bounds of 16 of these tiles. All processing in this study was performed on a per tile basis for each of these tiles over the 20-year dataset. The relatively high resolution of MAIAC gives it the ability to distinguish more fine scale features of AOD than the DB algorithm, described in the next section. Previous research has also shown that high resolution allows the retrieval of AOD over more complex geological features [11]. There is a land–sea detection mask contained within a parameter known as the Land Water Snow Ice Classification (LWSC), which is used in this work to mask ocean areas around Australia.

To perform its retrievals, MAIAC uses several fixed aerosol models; each of these has a look-up table. Over Australia, two aerosol models are used—these are 'Model 1', based on climatology seen in the east coast of the USA, with high summer humidities, and 'Model 2', which more closely resembles the more arid and dusty interior of the USA (for example, the coarse to fine mode AOD fraction is larger). This second model is used over the vast majority of the land whilst Model 1 is used in far northern and eastern regions [25]. The use of these two different models may cause differences in the AODs retrieved on the boundaries between these areas, as has been noted by Lyapustin et al. [25]. This has been found to be particularly prevalent in areas where there are large differences in absorption between two neighbouring models and when AOD levels are high.

#### 2.2.2. MODIS: DB AOD (MxD04 _L2)

The MODIS DB algorithm [26], so-named as it takes advantage of the 'deep blue' MODIS band (0.412 µm, near-UV), was developed to improve aerosol retrievals over bright surfaces, such as deserts. In the near-UV band, the land's surface reflectance appears lower than for the longer, visible wavelengths, allowing aerosol signals to show up more

clearly [26]. This algorithm utilises a hybrid approach in determining the surface reflectance contribution to the top of atmosphere radiance, consisting of a prescribed surface reflectance database and a dynamical method utilizing the Normalized Difference Vegetation Index. DB AOD is available in two separate datasets, one for each of Aqua and Terra, (MYD04_L2 and MOD04_L2, respectively) [26]. DB has a lower spatial resolution than MAIAC—roughly 10km at nadir, stretching to around 40km at swath edges. The data are presented as 'granules'—5 min segments of the swath overpass of the given area. Whilst spectral AOD is produced, this report utilises AOD at 550 nm for ease of comparison with MAIAC and because this is usually used as standard across research institutions for aerosol studies.

2.2.3. Aeronet

AERONET is a global system of ground based atmospheric monitoring stations which provide high temporal resolution data on a number of aerosol parameters, including spectral AOD, Ångström Exponent and Single Scattering Albedo [27]. AOD data from AERONET have a typical characteristic uncertainty of around 0.01–0.02 and are commonly considered to represent 'ground truth' [27,28]. Relevant data from all 18 AERONET stations (see Figure 1) based in Australia are obtained for this study; information regarding each of these sites is listed in Table 1.

AOD at 500 nm is available for 17 of these sites. In order to compare it to MODIS AOD, the AOD at 550 nm ($\tau_{550}$) is calculated from AERONET by interpolating the closest available AOD ($\tau_{500}$) using the Ångström exponent calculated from the nearest adjacent bands at 675 and 440 nm ($\alpha_{(675-440)}$), as in Equation (1). The Ångström exponents for various bands are available from the AERONET L2 data product and can be used to interpolate AODs at different nearby wavelengths [29], based on a linear fit. One site (Lucinda) presents AOD at 551 nm; this particular AOD is used in the analysis without alteration. Only the highest quality level 2 data are used, which have been quality assured and cloud screened.

$$\tau_{550} = \tau_{500} \left( \frac{\lambda_{550}}{\lambda_{500}} \right)^{-\alpha_{(675-440)}} \tag{1}$$

**Table 1.** Site Information for all 18 AERONET stations over Australia. The allocation of the land cover class associated with each site is described in Section 2.3.1.

| Site Name | Latitude | Longitude | Years of Data | Span | MAIAC Tile | Land Cover Class |
|---|---|---|---|---|---|---|
| ARM_Darwin | −12.4250 | 130.8910 | 2.8 | 2010–2015 | h30v10 | Medium Vegetation |
| Adelaide_Site_7 | −34.7251 | 138.6565 | 0.8 | 2006–2007 | h29v12 | Medium Vegetation |
| Birdsville | −25.8989 | 139.3460 | 6.9 | 2005–2019 | h30v11 | Bare |
| Brisbane-Uni_of_QLD | −27.4971 | 153.0136 | 1.6 | 2010–2015 | h31v11 | Mixed Urban |
| Canberra | −35.2713 | 149.1111 | 9.4 | 2003–2018 | h30v12 | Dense Vegetation |
| Coleambally | −34.8101 | 146.0644 | 0.8 | 2002–2003 | h29v12 * | Medium Vegetation |
| Darwin | −12.4240 | 130.8915 | 2.0 | 2004–2011 | h30v10 | Medium Vegetation |
| Fowlers_Gap | −31.0863 | 141.7008 | 3.1 | 2013–2018 | h30v12 | Sparse Vegetation |
| Jabiru | −12.6607 | 132.8931 | 9.5 | 2002–2019 | h30v10 | Medium Vegetation |
| Lake_Argyle | −16.1081 | 128.7485 | 10.0 | 2002–2020 | h30v10 | Sparse Vegetation |
| Lake_Lefroy | −31.2550 | 121.7050 | 4.6 | 2012–2020 | h28v12 | Medium Vegetation |
| Learmonth | −22.2407 | 114.0967 | 1.4 | 2017–2020 | h28v11 | Sparse Vegetation |
| Lucinda | −18.5198 | 146.3861 | 4.1 | 2014–2020 | h31v10 | Dense Vegetation |
| Merredin | −31.4931 | 118.2264 | 0.1 | 2006 | h28v12 | Medium Vegetation |
| Milyering | −22.0292 | 113.9231 | 0.1 | 2006 | h28v11 | Sparse Vegetation |
| Perth | −32.0081 | 115.8936 | 0.2 | 2005–2006 | h27v12 | Mixed Urban |
| Rottnest_Island | −32.0001 | 115.5017 | 1.0 | 2001–2004 | h27v12 | Mixed Urban |
| Tinga_Tingana | −28.9758 | 139.9909 | 4.6 | 2002–2012 | h30v11 | Bare |

\* Radii greater than 3 km from Coleambally station intersect the neighbouring tile, h30v12.

### 2.2.4. Auxiliary Data

Surface Classification

It is well known that the performance of AOD retrieval algorithms can be largely dependant on the underlying surface type (e.g., [17]). Aerosols can be distinguished more easily over dark surfaces (such as forests and oceans) than over highly reflective surfaces (such as deserts). For this reason, as part of our analysis, the performance of each AOD dataset is assessed as a function of land cover type. Figure 2 shows the land cover type around each AERONET site as characterised by the MODIS MCD12Q1 International Geosphere Biosphere Programme (IGBP) product. The IGBP product classifies the surface into 18 different types and is provided at yearly temporal resolution. In Figure 2, we show the average classification for 2016. It is apparent from Figure 2 that the heterogeneity around an individual site and between sites can be significant. For this reason, we simplify the surface classification according to the dominant land-cover type, as described in the next section.

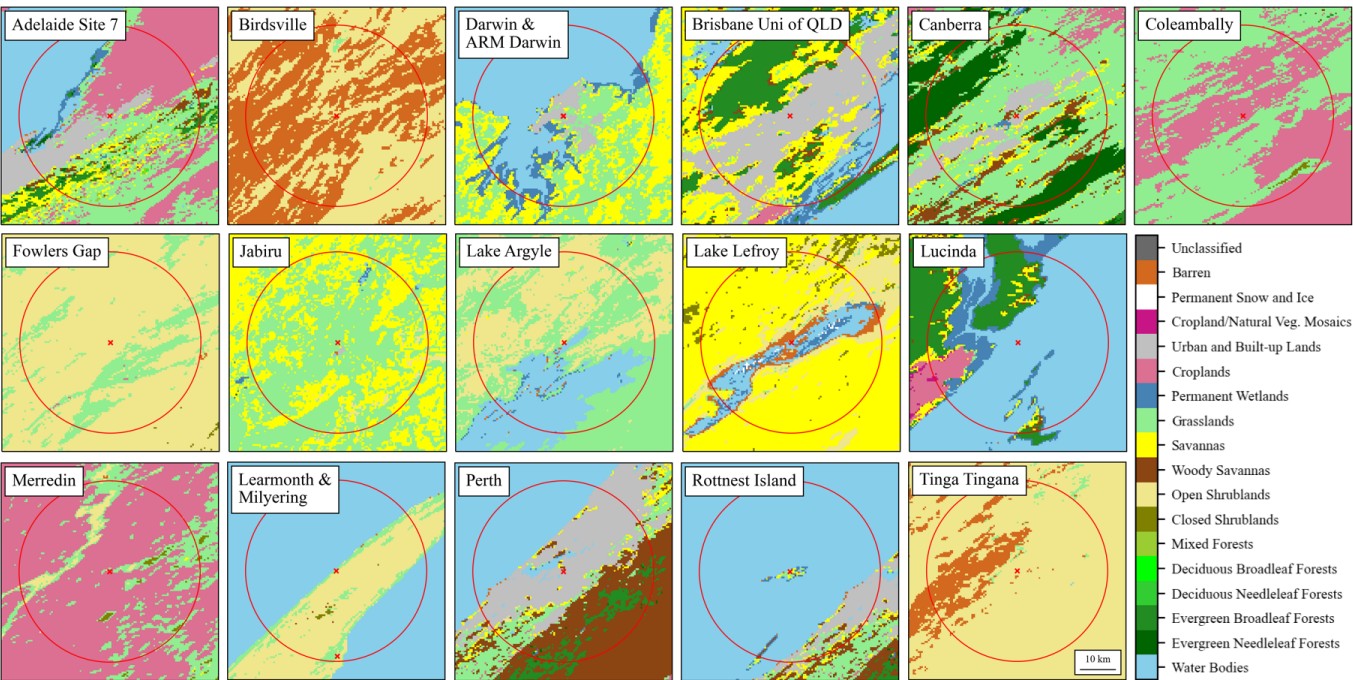

**Figure 2.** International Geosphere Biosphere Programme (IGBP) Land Surface Classification at 500 m resolution for each AERONET station (Year: 2016). Red circle is of radius 25 km around the sites.

CAMS Reanalysis AOD

To assist with the interpretation of the spatio-temporal distributions of the MODIS DB and MAIAC AODs, Copernicus Atmospheric Monitoring Service (CAMS) AOD reanalyses are employed [30]. These reanalyses, running from January 2003 to June 2021, provide 3-dimensional time-consistent atmospheric composition fields, including aerosols, chemical species and greenhouse gases [31]. They are produced using 4DVar data assimilation in Cycle 42r1 of the European Centre for Medium-Range Weather Forecasts' Integrated Forecasting System. We focus on the total and speciated AOD (at 550 nm), which involves the assimilation of either MODIS DT or DB or Advanced Along-Track Scanning Radiometer (AATSR) AOD retrievals. As such, we note that we might expect some dependency between the MODIS DB retrievals and the CAMS AOD reanalyses examined here.

Five aerosol types are included in the CAMS reanalysis AOD product: black carbon, dust, organic matter, sea salt and sulphate. Black carbon and organic matter tend to be produced from similar emissions sources, including biomass burning (such as wildfires) and pollution from fossil fuels and combustion, meaning they have both natural and anthropogenic origins [32]. Organic matter aerosols are also produced from other biogenic



sources [31]. Dust aerosols typically originate from arid and semi-arid environments as a result of wind induced uplift but can be entrained and carried substantial distances depending on their size [8]. Sea salt aerosols are generally produced as a result of sea spray and wave breaking, and near surface winds again play a role both in their formation and circulation, driving them over nearby land surfaces [32]. Sulphates can have both natural and anthropogenic origins, being produced naturally by volcanoes but also commonly in Australia by mining operations and industrial urban centres [33]. Here, we make use of the total and speciated AODs to determine spatio-temporal distributions of specific aerosol types over Australia for the period 2003–2020.

*2.3. Methods*

2.3.1. Surface Type Classification

A simplified land-cover type classification for each AERONET site was determined by considering all pixels which fall within 25 km of the site location. For each site, this area is indicated by the red circle in Figure 2. To simplify the classification and allow the generation of more robust statistics when ground-truthing the MODIS AODs, we created new, broader categories to group all sites into one of five surface types, based on levels of vegetation cover and urban land cover. These categories were: no or extremely low levels of vegetation ('Bare'), sparsely vegetated areas ('Sparse Vegetation'), medium density vegetation areas including croplands ('Medium Vegetation'), densely vegetated areas ('Dense Vegetation') and areas with considerable urban land coverage ('Mixed Urban'). Table 2 lists the thresholds used to determine the new surface classification. Of all the categories, the 'Mixed Urban' classification is the least stringent. This is deliberate and accounts for the fact that urban coverage tends to be relatively small but can be responsible for disproportionately high aerosol emissions and hence optical depths. Of the sites considered, the site with the highest IGBP urban classification is Rottnest Island at 60%, followed by Perth at 47%.

**Table 2.** Land Cover classification criteria. If the IGBP classification shows Mixed Urban coverage of >30 % and also satisfies one of the other criterion, the Mixed Urban classification takes precedence.

| Category | Technical Criteria |
|---|---|
| Bare | >50% Barren OR<br>>40% Barren & >40% Open Shrubland |
| Sparse | >50% Open Shrubland |
| Medium | >50% (combined) of any medium density vegetation (Cropland, Grassland, Closed Shrubland, Permanent Wetland, or Savanna) |
| Dense | >50% (combined) any Forest type OR<br>>25% (combined) any Forest type & >25% Other medium or dense vegetation |
| Urban | >30% Urban |

We note that whilst the IGBP land surface categories are generally stable on the macro-scale, around some sites there is significant inter-annual variability. To account for this, the average percentage cover for each of the original IGBP categories over a time span coinciding with the AERONET operating lifetime for each site is used to perform the simplified classification. Figure 3 shows the spatial distribution of land surface categories taken from the IGBP classification, compared with the new classification described above. The percentage of land taken up by each category is as follows: 52.6% Sparse, 40.8% Medium, 5.2% Dense, 1.2% Bare and 0.2% Mixed Urban. Visual inspection of the resultant simplified category map shows that the categories appropriately classify the land surface cover in a sensible way.

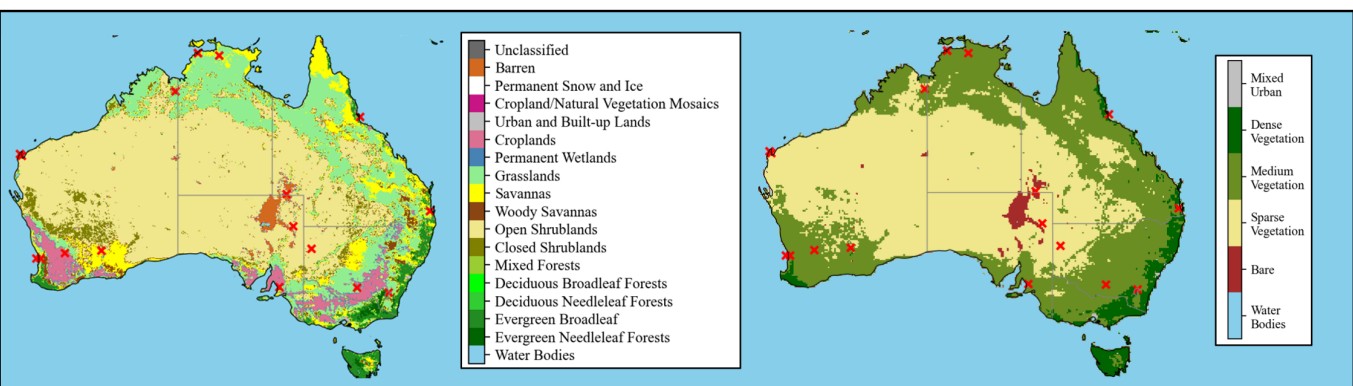

**Figure 3.** IGBP categories (**left**) and newly defined, simplified categories (**right**), with the location of AERONET stations indicted by red crosses.

### 2.3.2. Spatio-Temporal Co-Location of AERONET and MODIS AOD and Evaluation Metrics

Spatio-temporal co-location of AERONET and MODIS AODs was performed to enable validation of the MODIS data against the AERONET 'ground truth' data. The MODIS record from 2001 to 2020 was co-located against all available AERONET stations in Australia (see Table 1). Although AERONET can, in principle, provide measurements continuously throughout sunlit hours, Aqua and Terra typically only see any given AERONET station in Australia up to 4 or 5 times a day. Therefore, AOD observations from AERONET and MODIS were matched to each other when readings were within a ±30 min time window and 0.3 degree radius of each other. This window span has been commonly used in past research to collocate MODIS to AERONET AOD [28,34]. To achieve this, AERONET readings were averaged over a 1 h window (±30 min) centred on the satellite-station overpass time. A shorter 30 min window was also considered, but the results from this analysis showed no significant deviations from those obtained using the 1 h match-ups, which were thus chosen in order to give more robust statistics.

For the spatial segment of the co-location, AOD were first regridded to a common 0.1° by 0.1° grid. Subsequently, any grid box whose centre fell within ±0.3° of the AERONET station location was included in an overall average. The 0.3° radius was chosen to allow sufficient DB pixels to be included for a robust statistical analysis. Several other radii were tested using MAIAC over the range 1 km to 50 km (approx. 0.01° to 0.5°), and this showed that the dependence of key statistics such as the R, RMSE and Bias (Equations (2)–(4)) on radius typically flattened at values above 15 km (~0.15 deg). We note that for some sites, this behaviour was not seen: for example, at low radii, the MAIAC retrievals tended to be biased high over Lake Lefroy. The fact that the site is located on a large salt pan points to a misidentification of the surface as elevated aerosol loading, an error which is compensated by retrievals further from the site. Note that the difficulty of correctly classifying surface conditions over this location is also highlighted by Figure 2, where several points are identified as snow or ice covered.

We use a number of standard metrics to evaluate and compare the performances of each algorithm in this work. To evaluate the performance of each algorithm, in addition to *R*, *RMSE* and *Bias*, the *EE*, Equation (5), is also calculated.

$$R = \frac{\sum_i (x_i - \bar{x})(y_i - \bar{y})}{\sqrt{\sum_i (x_i - \bar{x})^2}\sqrt{\sum_i (y_i - \bar{y})^2}} \tag{2}$$

$$RMSE = \sqrt{\frac{\sum_{i=1}^{N}(AOD_{Satellite} - AOD_{AERONET})^2}{N}} \tag{3}$$

$$Bias = \overline{AOD}_{Satellite} - \overline{AOD}_{AERONET} \tag{4}$$

$$EE = \pm(0.05 + 0.15AOD) \tag{5}$$

The *EE* is commonly used in MODIS validation studies (e.g., [12,16,34,35]). It is designed to account for errors which are normally expected in a satellite retrieval of AOD. At low AOD, the error is usually dominated by both the instrument calibration and the surface reflectance estimations; this type of error is absolute and does not change with AOD. However, as AOD increases, assumptions within the aerosol model employed become much more significant, dominating at high AOD. The form of Equation (5) was defined so as to combine these two factors and has been widely used in satellite AOD retrieval validation studies of the type described in this work [12,16,35]. Typically, the assumption is made that if 66% of data points are found to lie within the EE envelope, a dataset can be considered validated or it can be considered that there is a 'good' match [36]. Clearly, a higher percentage of points falling within the EE envelope indicates a better performance. Through a global validation of DB data, Levy et al. [34] found the specific values used in Equation (5), which were chosen so that the envelope of ±*EE* values contained one standard deviation (i.e., ~68%) of the DB/AERONET matchups.

## 3. Results

### 3.1. Macro-Scale Spatio-Temporal Variation over Australia

To explore the internal consistency of the DB and MAIAC MODIS AOD records, the spatial distributions are evaluated at the seasonal timescale and Australia wide monthly mean temporal variations are examined. Because the datasets are produced with significantly different algorithms and assumptions, some variation is expected between them. For example, the higher spatial resolution of MAIAC allows it to pick out fine scale aerosol features that may be missed in DB [11].

### 3.2. Temporal Variations

Figure 4a shows the monthly mean time series of 'Australia mean' AOD at 550 nm (hereafter, 'AOD' will refer to the AOD at 550 nm specifically) from the DB and MAIAC algorithms. 'Australia mean' in this context means an average of all Australian land pixels and corresponds to the study area displayed in Figure 5. The deseasonalised variation for the whole record is also shown in Figure 4b. The time series was deseasonalised using the standard approach of creating a 20-year climatological monthly mean 'Australia mean' for each month and then subtracting this from the time series shown in Figure 4a. A trendline analysis was performed on the deseasonalised time series to determine the presence of upward or downward trends in AOD over the last two decades, and this is displayed in Figure 4c.

The variation in AOD with time suggests a seasonal cycle is present in the retrievals from both algorithms, with AODs typically peaking from November to February and at a minimum between June and August. The timing of these peaks coincides with Australia's known high aerosol seasons, in which wildfires are prevalent in the North and Australia's interior becomes dustier [6,8,22]. The MAIAC algorithm exhibits increased levels of AOD at almost all times of year over DB, with an overall average bias over DB of 0.023 (or roughly 50% larger). The notable exceptions to this are months where extreme high AOD peaks occur, and DB rises up above the maximum MAIAC AOD levels. This can be seen around January 2003, December 2006 and January 2020. The largest spike (January 2020) coincides with a very active fire season in Australia [37]. Overall, the mean level of each time series is relatively stable. This is reflected by the trend analysis performed on each individual sensor and retrieval algorithm, which shows a maximum value of $0.00050 \pm 0.00012 \text{ yr}^{-1}$ for MAIAC Aqua, where the error estimate is indicative of the uncertainty in the linear fit. The corresponding trend from DB Aqua is negative and slightly smaller, at $-0.00033 \pm 0.00017 \text{ yr}^{-1}$, in agreement with the value derived by Yang et al. [6] from 2002 to 2020 annual averages. For Terra, MAIAC shows a negative trend of $-0.00035 \pm 0.00011 \text{ yr}^{-1}$. All three of these linear fits imply a trend that is different from zero with *p*-values at or lower than 0.05. Conversely, the linear fit for DB Terra is not

statistically significant. Given these results, it cannot be definitively concluded that there is any overall trend in Australian AOD levels over the time period under consideration, particularly because the sign of the trend for a given satellite switches depending on the retrieval algorithm. Disagreement in the sign of the long term DB AOD trends from Terra and Aqua has been occasionally noted in the literature over megacities, for example in South America, for the period 2002–2010 [38]. Work by Provençal et al. [39] also finds no significant trend in AOD levels when focussing on three particular Australian cities for the period 2003–2015, from an analysis of the MERRA Aerosol Reanalysis dataset, which includes MODIS data assimilation.

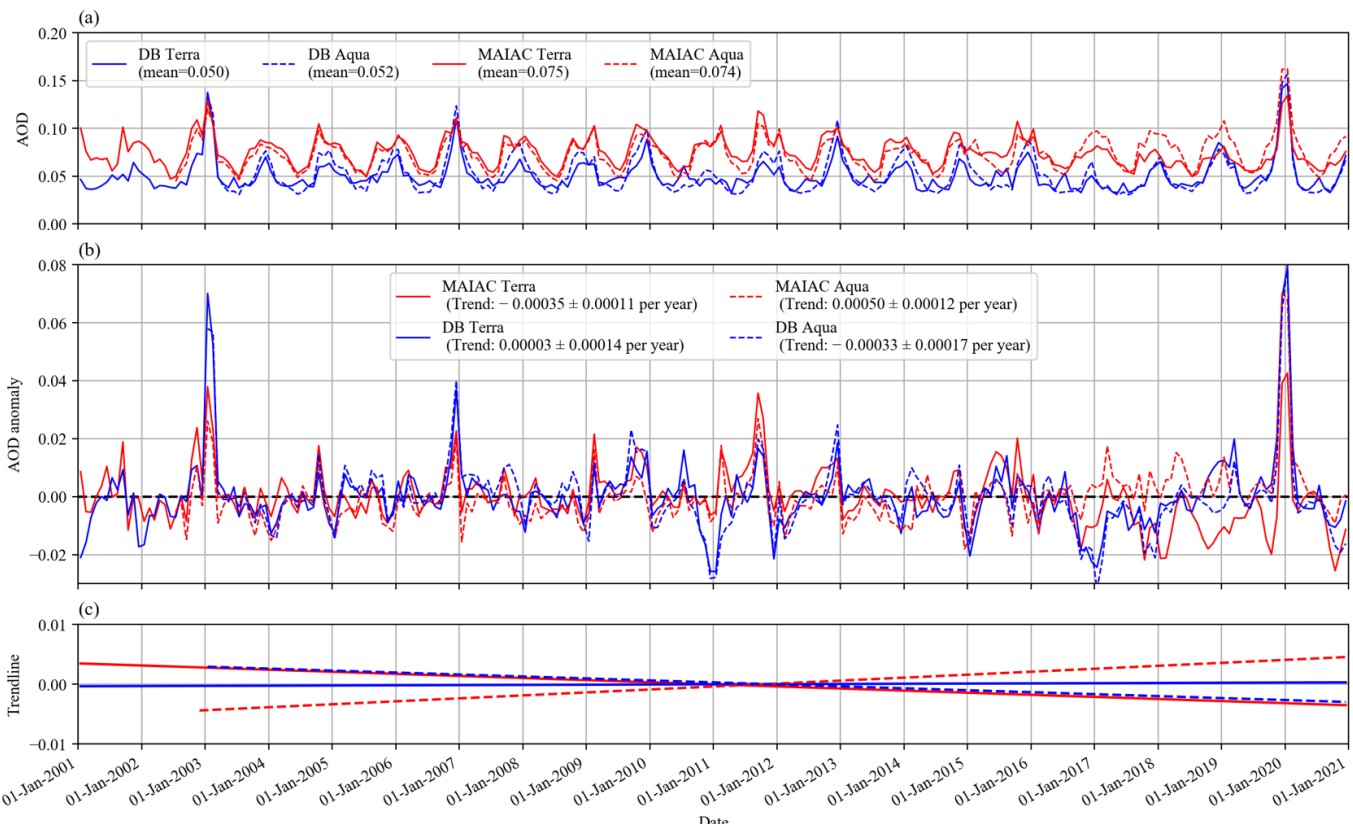

**Figure 4.** Monthly mean time series of Aerosol Optical Depth (AOD) for Deep Blue (DB) and Multi-Angle Implementation of Atmospheric Correction (MAIAC) over Australia (**a**), the deseasonalised time series, showing the AOD anomaly (**b**) and the fitted trendlines for these data (**c**), for all available land pixels.

One obvious feature from Figure 4 which influences the MAIAC trends is the deviation between MAIAC Terra and Aqua retrievals beginning in 2016 and continuing to the end of the time series. To our knowledge, this divergence has not been noted in the literature to date, and it would be worth exploring whether it is unique to Australia.

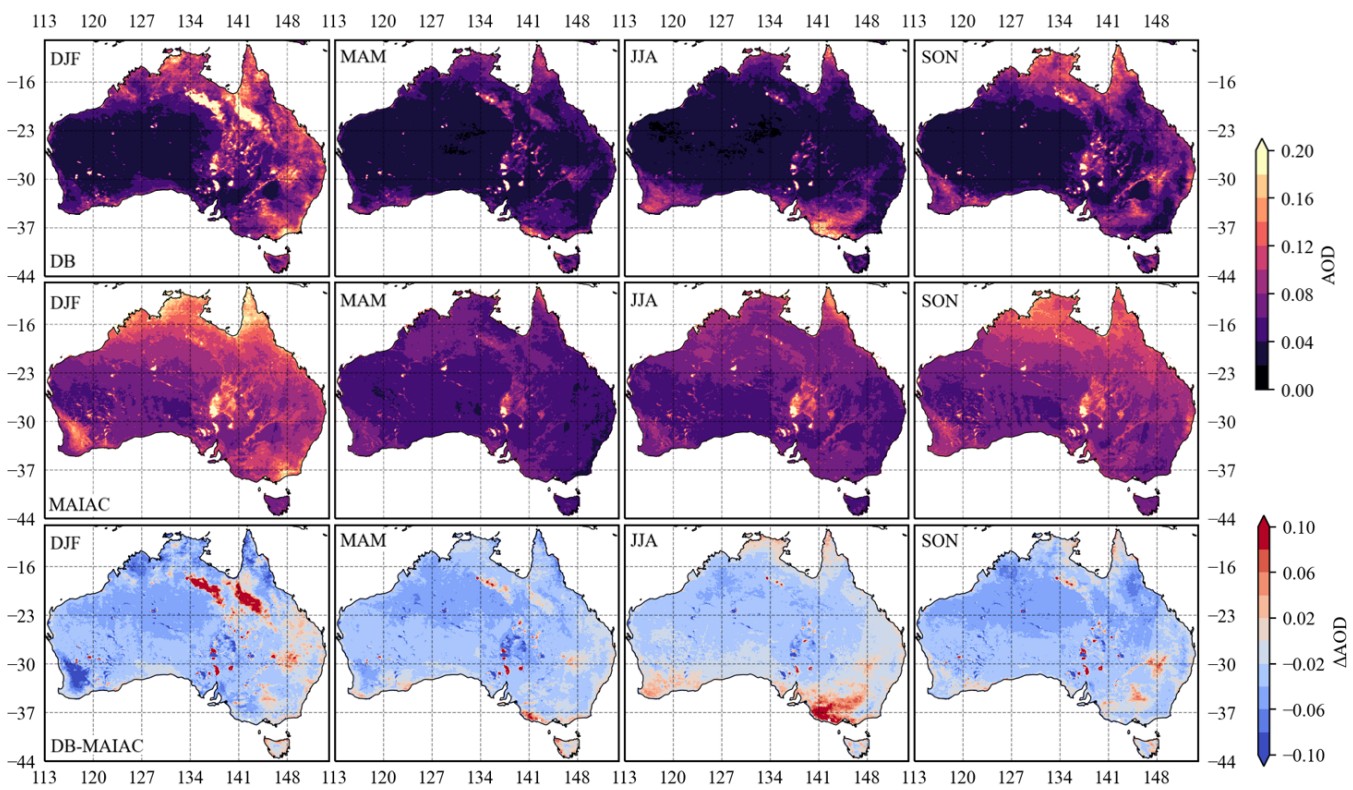

**Figure 5.** Seasonal spatial distributions of AOD over the period 2001–2020 for the DB (**top row**) and MAIAC (**middle row**) algorithms, and the algorithm AOD difference in the sense of DB-MAIAC (**bottom row**).

### 3.3. Seasonal Spatial Variations

To gain further insight into the seasonality shown in Figure 4, maps of the 20 year mean AOD distribution over Australia during each season were created. Figure 5 shows the spatial distribution by season for MAIAC and DB AOD (Terra and Aqua combined) and the difference between the algorithms in the sense of DB-MAIAC. It is evident from these images that, for the DB algorithm, low levels of AOD are found over the majority of the land surface in all seasons, with lowest levels seen in the Austral autumn and winter months (MAM, JJA), dipping to around 0.04. Higher levels of AOD are seen in the spring and summer (consistent with Figure 4), with northern areas recording average AODs that are around 3 times higher than those seen in the winter months. There are similar elevated AODs seen in the southwest in the winter and spring (JJA, SON), centred at approx. 118°E, 33°S and in summer (DJF) in the southeast (centred at approx. 148°E, 37°S). The southwest regions encompass large areas of cropland, and the southeast regions are a combination of forested and urban areas, with some cropland regions slightly more inland (see Figure 3). In JJA, there is a concentrated area of elevated AOD just west of Melbourne (141°E, 37°S), over a large area of mixed vegetation (croplands, forest and grasslands). A number of concentrated high AOD areas are seen inland (centred at approx. 137°E, 30°S) near a collection of ephemeral lakes including Lake Eyre-Kati Thanda and the Strzelecki Desert. These areas are known to be persistent sources of dust aerosol and appear active throughout the year. The top row in Figure 5 confirms previous findings by both Yang et al. [6] and Che et al. [10], who display the seasonal spatial distributions of the DB algorithm in a similar way and achieve similar results, albeit for slightly different time spans (2002–2020 and 1999–2021, respectively).

The middle row of Figure 5 shows that MAIAC exhibits a broadly similar spatial pattern to DB, with AOD levels becoming elevated in Austral spring (SON), leading to a peak in the summer (DJF) season. It appears that the dynamic range of the DB retrievals

is larger, with stronger peaks and typically lower minimum values across the Australian interior. This is confirmed by the bottom row of the figure, which shows the difference between the seasonally averaged AODs. Over the majority of the country and throughout the year there is a small negative bias between MAIAC and DB, with the former recording higher values consistent with the offset seen in Figure 4. However, DB shows two coherent features between 134 and 145°E, 16–23°S in DJF, that are much weaker in the MAIAC record. DB AODs over the Lake Eyre region also appear consistently higher (centred around 141°E, 30°S), and there is a region of elevated AOD centred on 141°E, 37°S in JJA that is not seen at all in the equivalent MAIAC retrievals. Conversely, the elevated AODs centred on 118°E, 32°S seen in the MAIAC record in DJF are not apparent in DB. Comparison of Figure 5 to Figure 3 suggests that the strong positive features in northern Australia during DJF occur over grassland. The marked negative difference in south-western Australia during DJF appears aligned with cropland surface cover. In some seasons for MAIAC, a vertical boundary artefact is present in the east of the country, a consequence of the aerosol model implementation in the MAIAC algorithm (see Section 2.2.1). This is most evident in summer, when AOD levels are generally higher. Caution should be used when interpreting AODs near this boundary.

In order to more confidently attribute the spatio-temporal patterns seen in Figures 4 and 5, we make use of the CAMS AOD reanalysis product. Although this is, by its nature, a model-observation hybrid, it brings together knowledge of the underlying meteorological and anthropogenic factors driving aerosol emission with quantitative estimates of total aerosol loading from satellite retrievals. Figure 6 shows the equivalent plot to Figure 4a, giving the monthly mean time series of the total CAMS AOD over the Australian land surface. Inspection of both figures indicates general agreement in the magnitude of the AOD, with CAMS total AOD tending to sit within the envelope of DB and MAIAC retrievals. The seasonality evident in the retrievals (Figure 4a) is also apparent in the CAMS record, although detailed comparison suggests that CAMS tends to agree more closely with DB in Austral autumn and winter, switching to better agreement with MAIAC in most Austral spring and summer seasons.

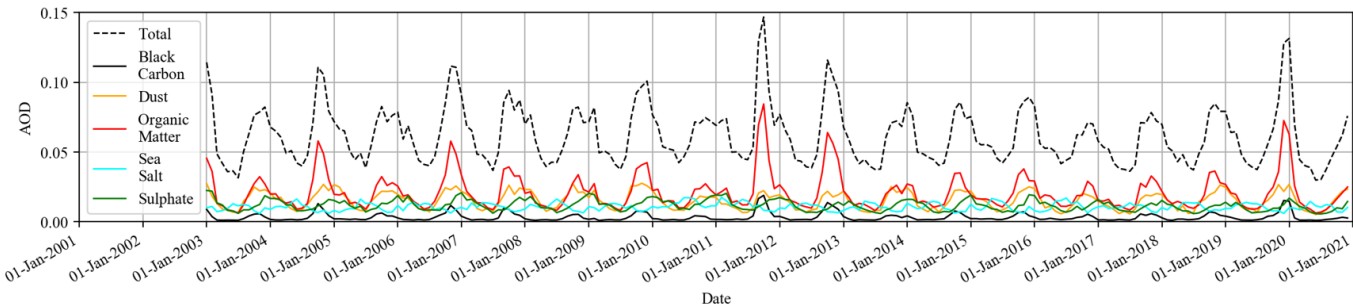

**Figure 6.** Monthly mean time series of speciated AOD types from the Copernicus Atmosphere Monitoring Service (CAMS) reanalysis over Australia for the period 2003–2020. This mean is the average AOD for each month over the entire Australian land mass, including Tazmania.

Figure 6 also shows how the total CAMS AOD is broken down into the various aerosol types. Looking at the speciated records, all aerosol types show a seasonal cycle which peaks broadly in tandem with the satellite data, as shown in Figure 4, with the exception of sea salt aerosol, which peaks around six months earlier. Within CAMS, Figure 6 indicates that the highest peaks in total AOD (in 2003, 2004, 2006, 2011, 2012 and 2019) occur as a result of spikes in the organic matter, and to a lesser extent, black carbon AOD. All of these peaks align with unusually high retrieved values for at least one of the MAIAC or DB algorithms, as seen in Figure 4. This analysis suggests that enhanced biomass burning activity during these periods is likely responsible for the large peaks in the observed records.

Analogous to Figure 5, Figure 7 shows the seasonally averaged CAMS AOD but separated into each aerosol type and covering the period 2003–2020. Panels (a–d) show

the smallest contributor to total AOD, black carbon aerosol, which shows very low levels year-round. Some seasonal dependency can be seen, with a small peak in the southwest (148°E, 37°S) in DJF situated between Melbourne and Canberra, coincident with large forested areas, which are prone to wildfires in the summer season. Some black carbon aerosol is also picked up in the far north in SON, also coincident with the wildfire season in that location, and which is largely covered by savannas and grasslands. Patterns in the organic matter AOD (i–l) tend to back up these inferences with regard to wildfire activity. High levels of organic matter AOD are seen over a more extensive area of the northern regions, particularly in DJF and SON. These features are consistent with the peaks seen in the same area from both MODIS algorithms (Figure 5), providing further evidence that the likely cause of these high AOD retrievals is biomass burning. The feature noted at 148°E, 37°S in DJF is also apparent in the organic matter AOD field. Moreover, elevated values are also seen in SON over the east coast. Again, both of these seasonal patterns are seen in the DB and MAIAC retrievals. Interestingly, organic matter seems to be the only viable cause of the aerosol peak seen in the MAIAC algorithm in DJF in the SW (centred around 116°E, 33°), with all other aerosol species showing a distinct lack of prevalence in that area.

Of the remaining aerosol types, a fairly strong seasonal dependency is seen for dust aerosols (Figure 7e–h) with the highest levels seen again in the Austral spring and summer. It is largely spread through the interior of the continent, with hotspots seen near Lake Eyre (141°E, 30°S). The CAMS analysis suggests that dust is the largest contributor to inland AOD in Australia. As might be anticipated, sea salt aerosol is elevated near the coastal areas and nearly entirely absent in the interior (m–p). High levels of this aerosol do ingress over land in the far north of Queensland (142°E, 44°S), peaking most strongly in the autumn and winter, in antiphase to the other aerosol types. Finally, over large swathes of the Australian interior, sulphate levels are very low (q–t), with the exception of the well-defined bullet-like peak in sulphate AOD at approx. 137°E, 20°S, evident in all seasons, which is coincident with the location of Mount Isa, a city in Queensland [40]. The sulphates here are a by-product of the mining industry in Mount Isa, home to mineral ores containing sulfur, which, upon processing, release large amounts of sulphur dioxide into the atmosphere, a precursor to sulphate aerosol [32,33]. The feature lines up well with the leftmost coherent feature captured by the DB algorithm (and to a lesser extent, MAIAC) around 134–141°E, 16–23°S. The seasonalities of these features also match, with the highest levels seen in DJF in both Figures 5 and 7. Figure 7 also shows small sulphate aerosol peaks along the southeast coast, near Sydney and Canberra, again which might be expected from a densely populated urban area and which is a consequence of human activity. This peak is strongest in DJF and weakest in JJA, which is reflective of average energy usage over those times [41].

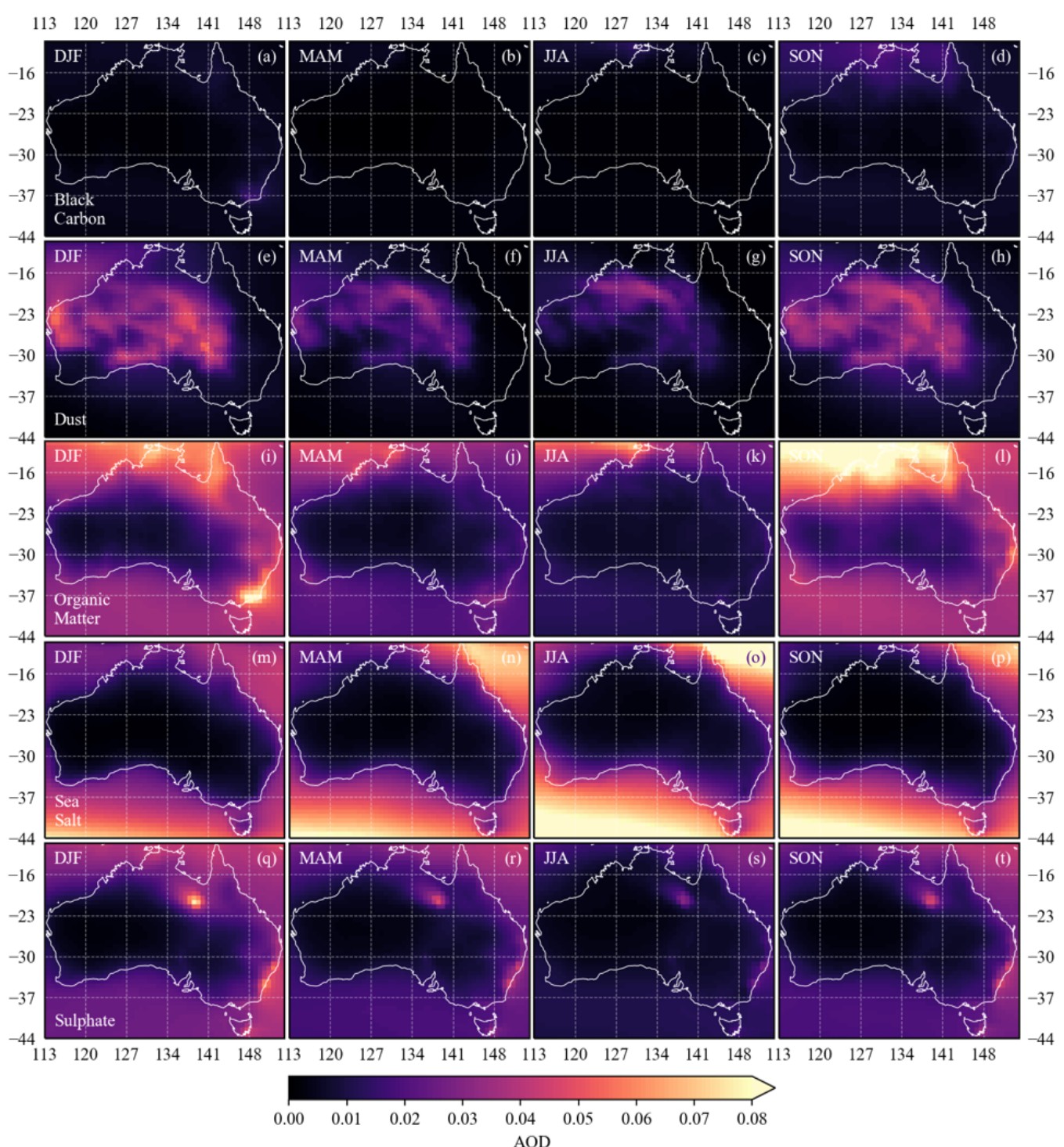

**Figure 7.** Seasonal spatial distributions of speciated AODs from the CAMS reanalysis over the period 2003–2020 for (**a–d**) Black Carbon, (**e–h**) Dust, (**i–l**) Organic Matter, (**m–p**) Sea Salt and (**q–t**) Sulphate AOD.

*3.4. MAIAC and DB Evaluation Using AERONET*

Although the results of Section 3.1 show that, broadly speaking, the seasonal and spatial behaviour of the MAIAC and DB datasets over Australia is similar, there are some notable discrepancies. To provide a further 'ground-truth' evaluation, in this section the satellite retrievals are compared to all 18 available AERONET sites across Australia, using the methodology outlined in Section 2.3.2. Figure 8 shows the results of this comparison.

In order for a fair comparison to be made, the MAIAC and DB datasets were sub-sampled to contain only pixels where both algorithms successfully made a retrieval, resulting in a total of 24,041 co-locations.

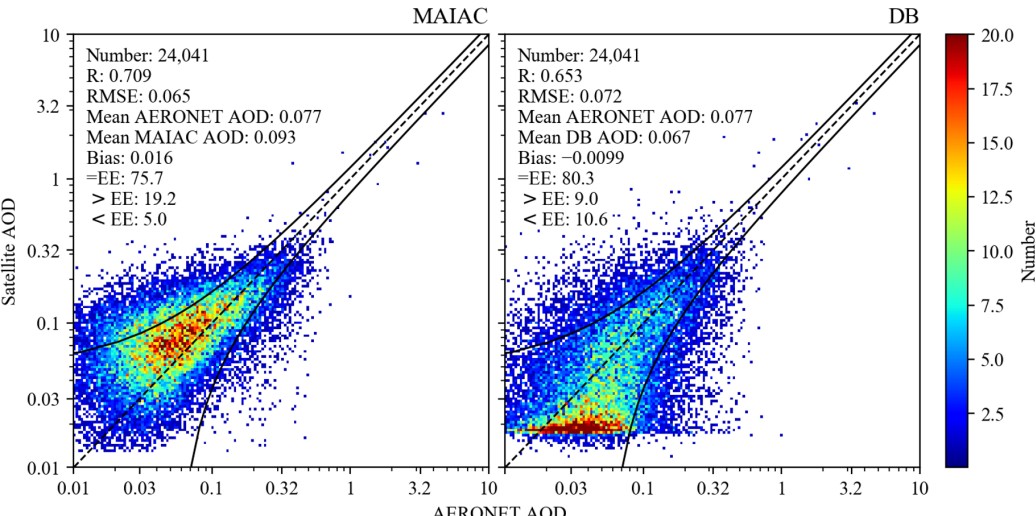

**Figure 8.** Results of the MAIAC and DB co-location to ground truth AERONET sites as a density scatter plot. The dashed black line is the 1:1 line, and the solid black lines are the upper and lower limits of the EE envelope. Statistics are printed on the graph representing the number of co-location matchups (Number), Correlation Coefficient (R), root-mean-square error (RMSE), Mean AOD, Bias and the percentage of points falling inside (=EE), above (>EE) and below (<EE) the Expected Error (EE) envelope. Note the log scale.

Broadly speaking, the retrievals from both algorithms show what would be classified as 'good' performance, with over 75% of MAIAC retrievals falling within the EE envelope, rising to in excess of 80% for DB. Over the whole distribution, the mean AERONET AOD is relatively low at 0.077, with the DB retrievals biased slightly low and the MAIAC retrievals slightly high. This relative performance is consistent with the behaviour seen in Figure 4 and implies that the true Australia mean AOD lies somewhere between the red and blue lines shown there. Both RMSE and R values hint at a slightly improved performance for MAIAC compared to DB, although the difference in both cases is relatively minor. It is notable that the contours in Figure 8 are rather more aligned with the 1-1 line for MAIAC, with the DB retrieval appearing to flatten out at an AOD value of around 0.02, perhaps indicating a minimum sensitivity issue and potentially limiting the usability of the data at AODs below around 0.03. Whilst the 'flattening-out' effect of DB retrievals at low AOD has not been noted previously in the literature, some publications show figures which hint at similar behaviour, e.g., Tao et al. [19] Figure 7 and Osgouei et al. [42] Figure 3; however, this has not been specifically commented on before.

Figure 9 shows the bias distributions (in the sense MODIS-AERONET) for all collocated observations for DB and MAIAC. The retrievals are also split according to the satellite platform. Figure 9 reinforces the messages from Figures 4 and 8 in terms of the overall bias between DB and MAIAC retrievals. In addition, the DB algorithm applied to Terra observations shows a slightly larger negative bias than DB Aqua observations when compared to AERONET. Mhawish et al. [11] found similar behaviour for DB over South Asia, in which DB Terra is slightly more negatively biased than DB Aqua. They postulate that differences in the bias for Terra and Aqua could be caused by instrument calibration. Over the same period, the MAIAC performance across both satellites appears more stable. However, it should be noted that decomposing the figure into pre and post 2016 highlights the offset noted in Figure 4 for the later period. This implies some form of instrumental artefact rather than a real diurnal difference in AOD is responsible.

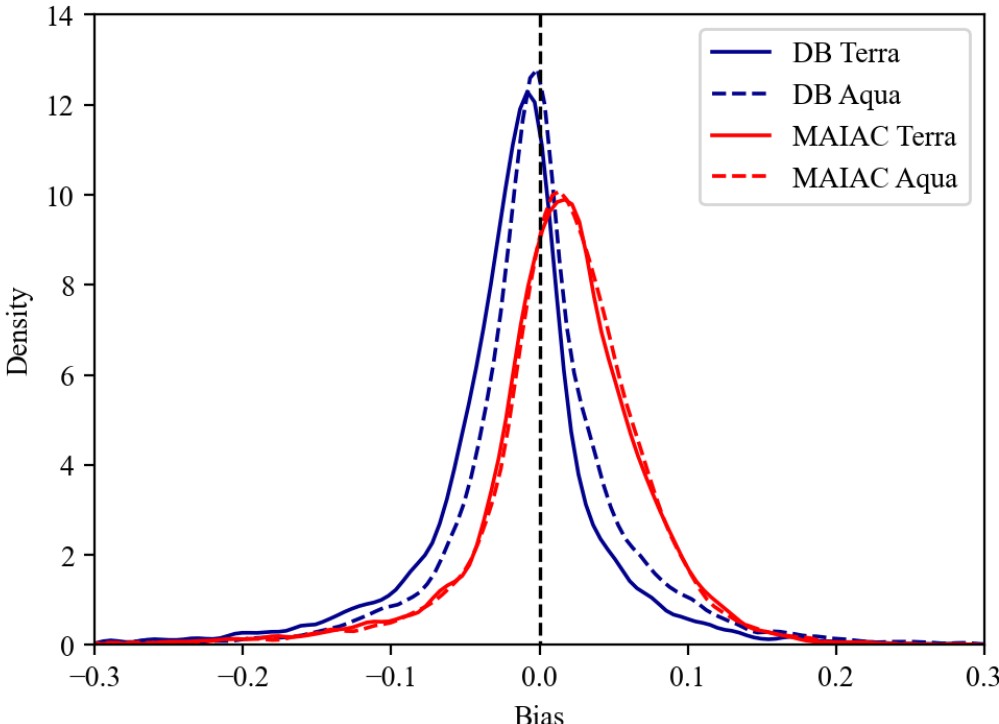

**Figure 9.** Bias distribution for MAIAC and DB over all Australian AERONET sites, separated into Terra and Aqua components.

For completeness, Figure 10 shows how the satellite bias varies as a function of AERONET AOD. It shows that both MAIAC and DB increasingly underestimate AOD as the AERONET AOD level increases. The figure shows that nearly all of the positive bias in MAIAC stems from co-location pairs in the first four AERONET AOD bins (which contain the vast majority of the data points). This shows MAIAC is overestimating low AOD for much of the time. Above an AOD of around 0.15, both algorithms experience mostly negative bias, which becomes increasingly extreme as AERONET AOD increases. Above an AERONET AOD of 0.25, the median and mean begin to fall below the bounds of the expected error envelope for both MAIAC and DB, although interpretation of the bias behaviour at these AODs and above must be performed with caution due to the limited number of collocations.

It is very likely that limited AERONET sampling in Australia will have some bearing on these results. In particular, the underlying surface type and/or its complexity might be expected to introduce specific biases or limitations in the quality of the retrievals. Therefore, in the next section we evaluate the level of agreement between AERONET and the MODIS retrievals as a function of surface type to give more insight into the strengths and weaknesses of each algorithm.

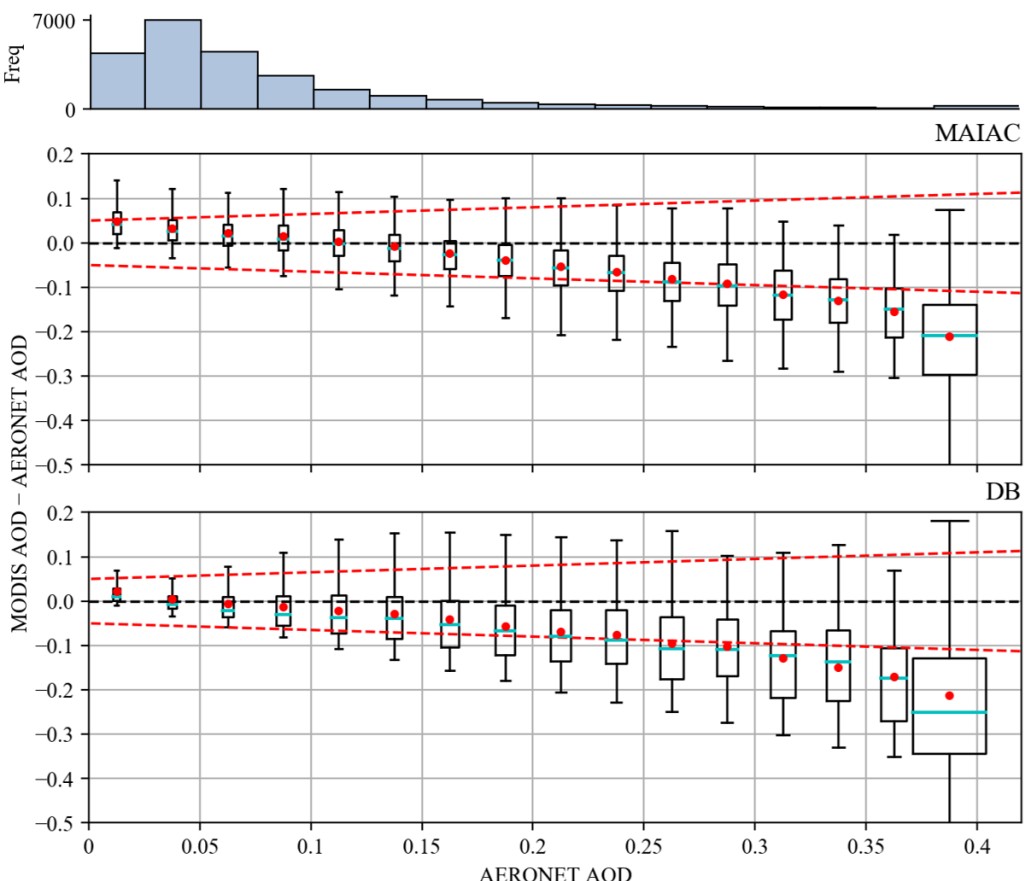

**Figure 10.** Box plots showing the MODIS-AERONET bias as a function of AERONET AOD for MAIAC (**top**) and DB (**bottom**). AERONET AOD was grouped into bins of 0.025. Plotted for each bin is the algorithms median and mean AOD value (blue line and red dot, respectively) and the interquartile range of values (box). The widths of the boxplots are proportional to the standard deviation of the retrieved AOD values for that bin. Points lying greater than two standard deviations from the mean AERONET value are not shown (for clarity). The frequency histogram shows how many co-location pairs make up each bin. The zero bias line is indicated by the black dashed line at x = 0 and the EE envelope is shown with the red dashed line.

*3.5. Spatio-Temporal Co-Location Results by Surface Type*

Figure 11 summarises R and RMSE for all the AERONET stations in this study over their individual periods of operation. It is relevant to note that several of the poorly performing sites with low R (<0.5) and high RMSE (>0.05), in particular on the west and south coast of Australia, are sites which had very short periods of operation of between 1 month and 2 years, and subsequently, data used to create these statistics contained very few observation pairs (generally < 200). For example, the Rottnest Island site had only 29 co-location pairs. Nonetheless, there appears to be no consistent relationship between record length and either R or RMSE across all of the sites, with a more obvious clustering being related to location and, potentially, surface type.

With this result in mind, Figure 12 shows density scatter plots of AERONET AOD against MAIAC and DB retrievals grouped according to the site surface type, using the classification outlined in Section 2.3.1. For MAIAC, R, RMSE and bias statistics are relatively consistent across all surface types, excepting 'Bare', which shows a markedly reduced R and enhanced high bias, particularly at low AODs. This is also reflected in the reduced number of points lying within the expected error envelope (<50%) for this surface type compared to >80% for the four other surface classifications. For DB, the correlation is also markedly lower for sites classified as Bare, but the distribution is more centred around the 1-1 line such that a greater percentage of points fall within the expected error. The values

of R are always smaller for DB than for MAIAC, with an accompanying small increase in RMSE for all but the Bare surface type. The flattening out at very low AOD apparent in Figure 8 is most obvious for 'Dense' vegetation classification, although there is a clear minimum retrieved value in most cases.

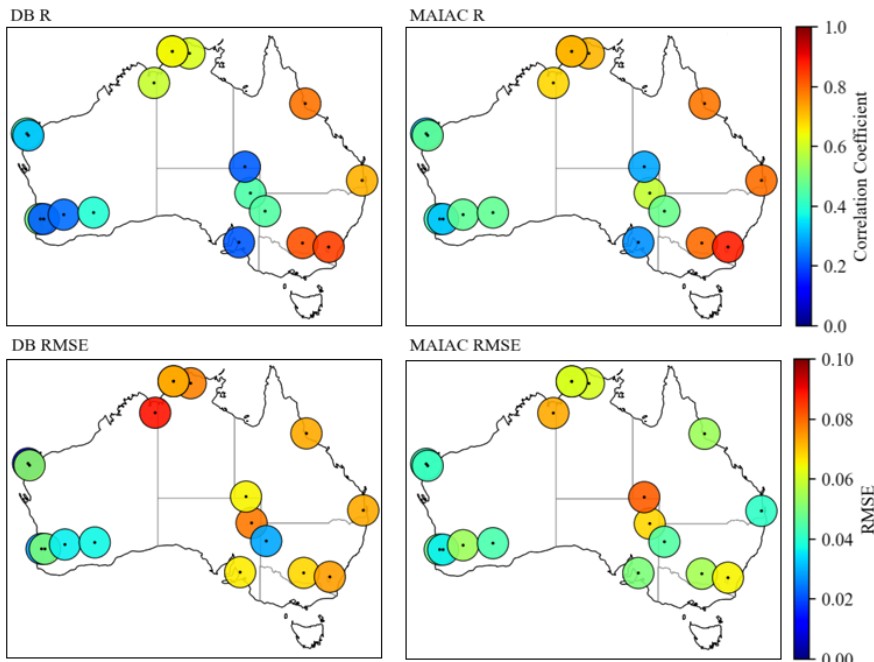

**Figure 11.** Correlations Coefficients (R, **top**) and RMSE (**bottom**) between satellite data (DB left and MAIAC right) and each AERONET station over Australia for collocations in the period 2001–2020. The operational period of individual AERONET sites is highly variable, extending from 1 month to 10+ years (refer to Table 1).

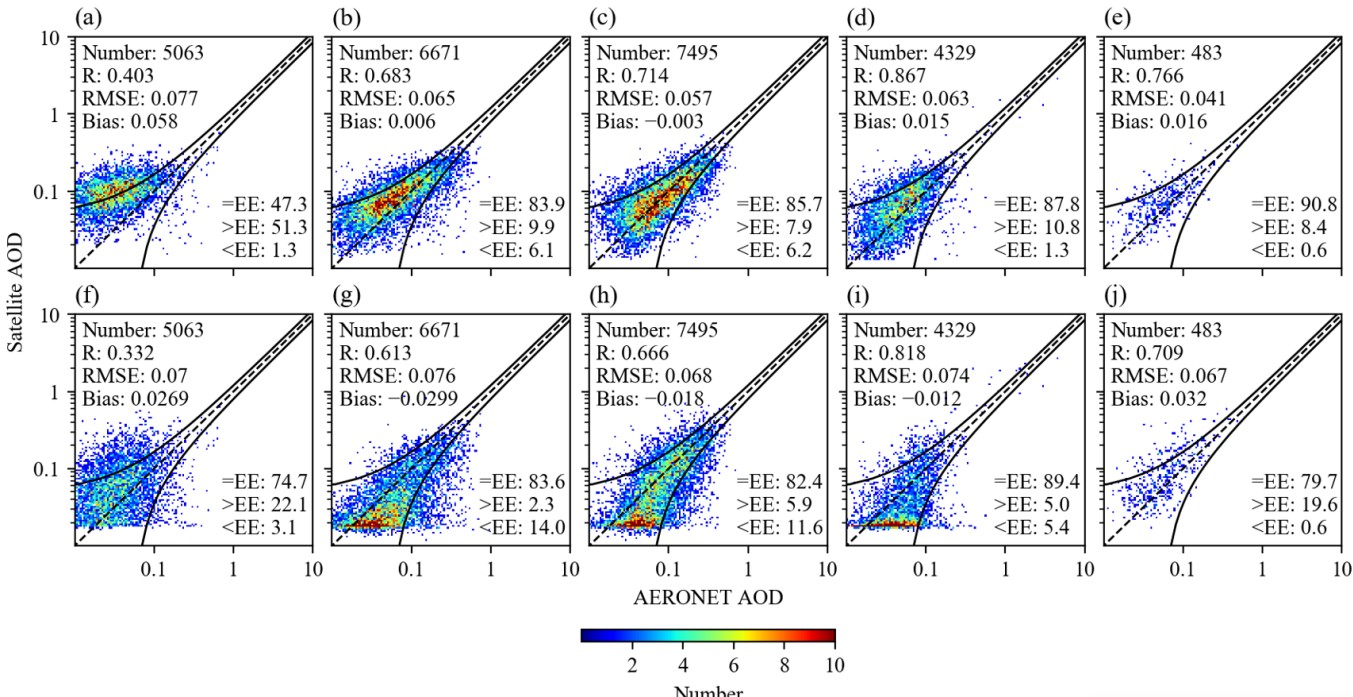

**Figure 12.** Results of the MAIAC (**a**–**e**) and DB (**f**–**j**) co-location to AERONET ground truth as density scatter plots and presented as a function of surface type, where: (**a**,**f**) Bare, (**b**,**g**) Sparse, (**c**,**h**) Medium, (**d**,**i**) Dense and (**e**,**j**) Mixed Urban. All stats and line indicators are as presented in Figure 8.

A summary of the performance of each algorithm over the different surface classifications is provided by Figure 13. The figure reiterates the very good agreement seen in the MAIAC retrievals across vegetated and urban categories, with the algorithm's performance over Bare surfaces a clear outlier. DB performs more consistently across all five surface classifications but shows larger biases in terms of both the mean and median across the majority of surface type classifications considered here.

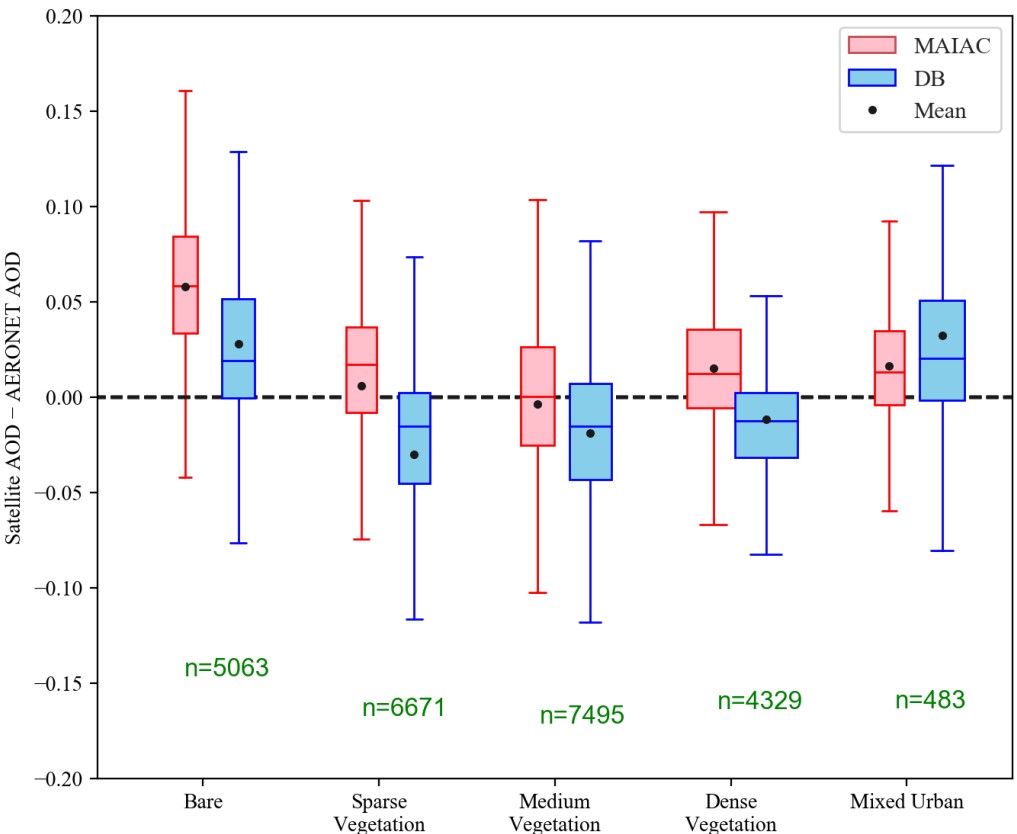

**Figure 13.** Box and whisker plots illustrating the bias variation as a function of surface type around the AERONET sites. Green text indicates the number of co-location pairs per category. Widths of the boxes are proportional to the standard deviations, and the central horizontal lines mark the medians. Fifty percent of data falls within the bounds of each box. Outliers have been removed—lower limit is the AOD value of the lower limit of the inter-quartile range (IQR) subtract 1.5IQR. Similarly, upper limit is upper quartile plus 1.5IQR.

## 4. Discussion

We evaluated the performance of the MODIS MAIAC and DB AOD retrieval algorithms over the Australian continent by co-locating the satellite observations with the AERONET system of ground-based sun photometers. The performance was further assessed as a function of aerosol loading and also by surface type to elucidate whether the algorithms had particular problems retrieving over a given surface. All possible AERONET sites and two decades of AOD data were analysed to perform these assessments. Unfortunately, the small number of AERONET sites in Australia, many of which operated for a limited duration, limits how far the accuracy assessment can be taken. Very few of these sites are placed in inland regions, with the vast majority in or near coastal areas, in places which are not typical of the majority of Australian land and may not have typical aerosol profiles. More aerosol data are needed from the interior of the region to further assess the performance of these satellite retrievals. A related limitation of this study is that the simplified land cover classification is somewhat subjective and based on perceived 'thickness' of vegetation cover. This could affect the interpretation of the accuracy of algorithms

over different surface types. Future assessments may explore whether results significantly deviate based on changes in the classification system.

## 5. Conclusions

This article makes a new contribution to the accuracy assessment of both the DB, and in particular, the MAIAC AOD retrieval algorithms over Australia. We have performed a detailed comparison with all available AERONET stations covering the period 2001 to 2020. Spatio-temporal patterns of AOD from both algorithms were also investigated on a seasonal and monthly basis, and the CAMS reanalysis dataset was used to gain insight into aerosol types and sources. Our main conclusions are as follows:

1.  *Temporal Analysis:* A seasonal cycle of AOD was found to be present over Australia, which both the DB and MAIAC algorithms pick up. The AOD levels peak in the Austral spring and summer, also confirming findings made by Yang et al. [6] and Che et al. [10] for the DB algorithm. At almost all times of year, MAIAC displayed monthly averaged AOD levels, which were around 50% higher than that of DB for both satellites. Exceptions to this occur when there are large peaks in the AOD record. Analysis of CAMS reanalysis data shows a clear association of these elevated AODs with fire activity, suggesting that DB tends to overestimate smoke aerosol compared to MAIAC. Analysis of the long-term trends in the data shows very small values for both algorithms applied to both Terra and Aqua based sensors. Although the small negative trend derived from DB Aqua agrees with that quoted in previous work [6], results from DB Terra show no significant trend. Moreover, trends derived from MAIAC Aqua have inconsistent signs with those from DB Aqua. The trend from MAIAC also changes sign for the Terra platform. Given this, we cannot confidently assert that a trend exists in the AOD over Australia over the past two decades. It was also found that the deviation between MAIAC Terra and Aqua AODs increased in the period beginning in 2016. This has not been noted before in the literature, and it would be interesting to know whether this deviation is also apparent in other regions.

2.  *Spatial analysis:* The seasonally averaged spatial distributions of AOD for both the MAIAC and DB algorithms were generally consistent. Over large swaths of Australia, both algorithms retrieved very low average AOD, in all seasons, though values are higher for MAIAC. This spatial analysis also revealed differences in AOD peak areas between the two algorithms. Both showed a very spatially heterogeneous distribution of AOD in all seasons, with higher levels of AOD in the northern and eastern regions, which is particularly prominent in peak seasons (summer and spring). The MAIAC algorithm also shows strong peaks in AOD in the south-western regions in DJF, in areas covered in cropland.

3.  *Performance against ground sites: (a) Overall* Whilst both sites exhibit good performance overall, MAIAC was found to perform generally slightly better than the DB algorithm in almost all areas when compared to the ground truth stations. Over key evaluation metrics, MAIAC (R = 0.709, RMSE = 0.065) outperforms DB (R = 0.0653, RMSE = 0.072). We also find that MAIAC tends to be biased slightly high, whilst DB is biased slightly low, with the magnitude of bias smaller for DB. The 'true' AOD level is hence likely to lie somewhere in between these retrievals. The typically higher values of R for the MAIAC retrievals are manifested in terms of the distribution of points along or offset from the 1-to-1 agreement line, as opposed to a less defined clustering in the equivalent DB values. We also find evidence of a distinct lack of sensitivity of the DB retrievals at very low AODs, with a noticeable 'flattening-out' in the retrieved values when AERONET AODs are less than around 0.3. Although there are hints of this in previous work, to our knowledge this is the first time it has been so evident. We postulate that it may be related either to a lack of sensitivity or possibly to the discretisation used in the DB algorithm.
    *(b) By Surface Type* The quality of AOD retrievals was found to vary based on the underlying surface type. Better performance was found for both DB and MAIAC

over Sparse, Medium and Dense vegetation cover, with the worst performance being seen over Mixed Urban and Bare surfaces. Mixed Urban and Bare surfaces make up only 0.2% and 1.2% of the Australian land mass, respectively, according to the simplified classification used here. Therefore, the performance over the other 98.6% of the land surface indicates that both algorithms are able to retrieve AOD with a good level of accuracy over the vast majority of the Australian surface, making both algorithms applicable to use in (large-scale) studies of Australian aerosol. Across all five surface classifications used here, MAIAC's advantage in terms of a slightly higher R is retained. MAIAC also shows uniformly lower RMSE values than DB except over bare surfaces. The larger RMSE in this case is related to the more marked positive bias that MAIAC displays in these locations.

This work has shown that both the MAIAC and DB algorithms are suitable for use in regional scale aerosol studies in Australia. However, our findings suggest MAIAC AOD may be a better option to use for such studies due to its higher spatial resolution, ability to pick out fine detail over complex features [16] and marginally better performance both overall and over most surface types compared to DB.

**Author Contributions:** Conceptualization, M.S. and H.B.; methodology, M.S. and H.B.; software, M.S.; validation, M.S. and H.B.; investigation, M.S. and H.B.; writing—original draft preparation, M.S.; writing—review and editing, M.S., H.B. and A.S.; visualization, M.S.; supervision, H.B. and A.S.; funding acquisition, H.B. and A.S. All authors have read and agreed to the published version of the manuscript.

**Funding:** This work was supported by the Grantham Institute—Climate Change and the Environment. H.B. was partially funded through NERC's support of the National Centre for Earth Observation under grant award no. NE/R016518/1. A.S. is supported by the Met Office Hadley Centre Climate Programme funded by BEIS and Defra.

**Data Availability Statement:** MODIS DB and MAIAC AOD data and IGBP land surface data are publicly available from the NASA Level-1 and Atmosphere Archive & Distribution System Distributed Active Archive Center (LAADS DAAC) at: https://ladsweb.modaps.eosdis.nasa.gov/, last accessed: 5 April 2022. AERONET AOD data are publicly available from the NASA Goddard Space Flight Centre (GSFC) webpage at: https://aeronet.gsfc.nasa.gov/, last accessed on 20 February 2022. CAMS AOD data are publicly available from the Copernicus Atmosphere Monitoring Service's global reanalysis (EAC4) monthly averaged fields product, at: https://ads.atmosphere.copernicus.eu, last accessed on 3 May 2022.

**Acknowledgments:** Thanks to the providers of aerosol and land surface data used in this study.

**Conflicts of Interest:** The authors declare no conflict of interest.

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
