# Peer review of "An Evaluation of Two Decades of Aerosol Optical Depth Retrievals from MODIS over Australia"

_remotesensing, doi:10.3390/rs14112664_

Round 1

Reviewer 1 Report

The present paper by Shaylor et al. adds another contribution to present evaluation studies of the MODIS DB  and MAIAC AOD retrievals. The added value of this paper is, that it is focussed on detailed regional evaluation over Australia, including analysis with respect to seasonality and underlying surface type. The paper is well-structured and concise, I recommend publication after some minor revisions as stated below.

* L339-L242: Interesting finding and definitely worth to investigate in more detail. It would be nice to add a note about also to the conclusion section.

* L460-466: Is the DB-AOD "flat out" at low AOD values a common limitation of this retrieval? If yes, a reference to previous studies should be given,  else this finding should also be noted (a little more specific than in L567) in the conclusion section.

* Fig.8: Labels of statistics in the panels are both 'Mean MAIAC AOD'

* Fig.10: It is stated in the caption, but please add MAIAC / DB labels to the appropriate panels.

Typos:

* L51: Austral

* L140: are the areas are

*L397: Figure 4 -> Figure 6

Reviewer 2 Report

Dear all,

The manuscript has evaluated the performance of the Multi-Angle Implementation of Atmospheric Correction (MAIAC) and Deep Blue (DB) retrieval algorithms for estimating Aerosol Optical Depth (AOD) from Moderate Resolution Imaging Spectroradiometer (MODIS) images  in Australia compared each other and against ground-based observations. It has well organised and it clearly demonstrates the performance of algorithms to estimate AOD in Australia. It reveals the better performance of MAIAC than DO. 

Some comments will be provided bellow: -
1- Line 196 and equations 1 (a) how the (a) is calculated, it is a constant or it is a coefficient. You could add extra information about it. 
2- Line 397 is a Figure 6, not Figure 4.
3- You could check the numbers of figures in the captions and in the text.
4- line 527 in the most Australian continent not (Australian continent). This is because some countries in the continent are not included in the study such as New Zealand and others.

Best regards,

Reviewer 3 Report

GENERAL COMMENTS

In this paper the authors present the analysis of the Aerosol Optical Depth Retrievals From MODIS Over Australia in the last two decades.

From the technical point of view the analysis of data is coherent with the objective, so I recommend publication with the following minor revision.

SPECIFIC COMMENTS

2.2. Data Sources 

Please specify the download site of data presented at:

2.2.1. MODIS: MAIAC AOD (MCD19A2) 

2.2.2. MODIS: DB AOD (MxD04_L2) 

2.2.3. AERONET 

2.2.4. Auxiliary Data  - CAMS reanalysis AOD 

Line 292 - Please explain directly the meaning of the parameter EE that is introduced in eq.5 without a sufficient explanation

Line 294 – “The EE is commonly used in MODIS validation studies” 

Please add references

3.2. Temporal Variations 

This chapter should be reformulated, explaining accurately   the methodology utilized for:

Deseasonal analysis and the Spatial average ??? it is over Australia ??? explain better ….

figure 4b is quite confusing, pls put trend-analysis in an additional panel

Line 319 : “A clear seasonal cycle of AOD is observed for both algorithms, ….”

It is not so clear …. 

Lines 319-342 : Please cite (if appropriate) the following previous works:

-       Alpert, P.; Shvainshtein, O.; Kishcha, P. AOD trends over megacities based on space monitoring using MODIS and MISR. Am. J. Clim. Chang. 20121, 117. [CrossRef]

-       Provençal, S.; Kishcha, P.; da Silva, A.M.; Elhacham, E.; Alpert, P. AOD distributions and trends of major aerosol species over a selection of the world’s most populated cities based on the 1st version of NASA’s MERRA Aerosol Reanalysis. Urban Clim. 201720, 168–191. 

Figure 6 : Is there any kind of spatial average ??? if YES provide details 

Line 397 – “Figure also shows how the total CAMS AOD ….” => Figure 6 ???

Figure 7 should be paneled a,b,c …… and the relative legend self-explaining
